# Which Spatial Elements Influence Waterfront Space Vitality the Most?—A Comparative Tracking Study of the Maozhou River Renewal Project in Shenzhen, China

**Yating Fan** [1] , **Da Kuang** [1] , **Wei Tu** [1] **and Yu Ye** [2,3,*]

1 School of Architecture and Urban Planning, Shenzhen University, Shenzhen 518060, China; fyt_fighting@szu.edu.cn (Y.F.); da.kuang@szu.edu.cn (D.K.); tuwei@szu.edu.cn (W.T.)

2 College of Architecture and Urban Planning, Tongji University, Shanghai 200092, China

3 Key Laboratory of Ecology and Energy-Saving Study of Dense Habitat (Tongji University), Ministry of Education, Shanghai 200092, China

* Correspondence: yye@tongji.edu.cn; Tel.: +86-182-1726-8257

**Abstract:** Urban waterfront renewal, especially public space improvement, is important for regaining waterfront space vitality. However, existing studies constrained by sparse and hard-to-access data are hard to explore how changes in spatial elements during waterfront renewal would affect space vitality. Waterfront space vitality comprises social vitality represented by public behaviors and economic vitality represented by urban functional facilities. Taking the Maozhou River renewal project in China as an example, we collect spatial elements and vitality on corresponding periods in 2018 and 2020 (before and after the renewal construction) and use multiple linear regression models to assess the relationships. We find that the functional diversity (e.g., commercial and cultural facilities) and design quality (e.g., path density and the shoreline's proximity to the water) are the two most influential spatial elements affecting space vitality during waterfront renewal. Overall, the use of two-time datasets has generated strong evidence for measuring waterfront revitalization.

**Keywords:** urban waterfront; vitality; spatial elements; spatio-temporal differentiation; multi-sourced urban data

## 1. Introduction

The urban waterfront has undergone a dramatic transformation from industrial-era prosperity to post-industrial-era decline and now to the present revitalization [1,2]. Owing to the remarkable resource conditions (e.g., livelihood, ecology, and transportation) and geographical locations of waterfront spaces, their renewal promotes vitality by creating a public gathering place, activating society, upgrading industries, and boosting the economy [3–5]. However, waterfront renewal may not always satisfy the demand for public and diverse functional activities [6] and may instead result in low utilization of the urban waterfront [7]. Such divergent findings highlight the need for further research. Here, we explore how changes in spatial elements (i.e., scale, design, accessibility, and function; see Section 3.2.1) influence space vitality (i.e., the density and quality of social vitality and economic vitality; see Section 3.2.2) before and after a waterfront renewal.

Based on environmental behavior theories, space vitality is often interpreted—from an urban sociological perspective—as describing the vibrancy felt by people [8–10]. It is recognized that space vitality can be improved through high-quality space design [11]. According to Jan Gehl [12], space vitality is reflected by non-survival-based social and leisure activities occurring in a space. More diverse urban functions can stimulate more urban life activities, which represent more dynamic space quality and vitality [13,14]. Thus, determining how the changes in spatial elements influence space vitality in a renewal project is important to guide planners and architects toward better vitality revitalization [15,16].

The study of spatial elements and space vitality used to rely on qualitative and traditional data such as observation and personal experience [17]. With the development of new geospatial tools and multi-sourced Big Data that allow increased accessibility, larger space range, and longer time span, the quantitative measurement of spatial elements and vitality change has become more precise [9,10,18]. For spatial elements, the built-space morphological characteristics can be analyzed by extracting information through computer algorithms from historical remote sensing images, drone aerial photos, and street view images [19–21]. For evaluating space vitality, represented by a large amount of public behavior and urban functional facilities, location-based spatio-temporal behavior data, map apps, and social media data have also been widely used [22–24]. Using the aforementioned data and methods, researchers have explored the association between the design quality of space and vitality in waterfront spaces through statistics [7,25].

However, although these tools and data help in exploring how the improvement of built environments influences space vitality, few studies focus on a waterfront renewal process by tracking data and measuring changes in its spatial elements and space vitality. This might be due to the limitation of data accuracy on the spatial scale and the difficulty of acquiring data in the available time span [26]. Therefore, we take a spatio-temporal differentiation perspective and track two-time datasets for corresponding periods before and after the waterfront renewal to measure changes in the spatial elements and vitality separately. Further, we apply multiple linear regressions to analyze the corresponding relationships to determine the most influential spatial elements for improving waterfront space vitality.

The remainder of this paper is organized as follows. Section 2 reviews the literature on waterfront renewal and space vitality. Section 3 describes the methodology. Section 4 discusses the quantitative results. Section 5 elaborates on the contributions and limitations of this study compared with the existing studies, followed by a conclusion in Section 6.

## 2. Related Studies

### 2.1. Waterfront Space Vitality Measurement Based on Multi-Sourced Data

Because of the traditional observation limitation in complex urban problems [17], many studies recently started using spatio-temporal Big Data to measure vitality in public spaces such as streets [8,27], pergolas [28], and waterfront spaces [29,30], which are all linear urban public spaces. Waterfront public space vitality is revealed by human activities, including public behaviors (social vitality) and urban functional facilities (economic vitality) [3,31]. For social vitality, some characteristic indicators of public behaviors are used, such as space revisit rate [32], linear and surface density of space activity [7,25,33], and user diversity [7,34,35]. These data for public behaviors can be obtained from long-term data containing users' attributes and their spatio-temporal information, as tracked by location service products such as cellular data transmitted by cell towers [18], location-based services based on GPS [23], and wearable sensors [36]. For economic vitality, point of interest (POI) and social media attendee data are also used to explore semantic information on business, cultural, and leisure activities, which represent the intensity and diversity of economic functional facilities [22,24,29,30].

### 2.2. Quantitative Analysis of Waterfront Spatial Elements

Recently, the close relationship between the built environment's spatial elements and space vitality in it has been increasingly confirmed by scholars using quantitative research methods [18,37]. Ewing and Cervero [38] proposed the built environment's quality and morphological characteristics classified into five dimensions (density, diversity, design, destination accessibility, and distance to transit) as assessment frameworks for evaluating the influences of the built environment [27,28]. Several agreed-upon design elements are used as standard principles for a high-quality waterfront space, including ecological landscape [4,39], public function [3,6,40], traffic and pedestrian lane [5,41], and historical and cultural [5,42] elements. Furthermore, based on fruitful qualitative research, scholars

recently began to quantify waterfront spatial elements according to the spatial location, which is mostly divided into three categories: "waterbody–shoreline–inland" [16,43]. Waterbody ecological elements such as river width, revetment type, and water cycle [39,43]; spatial elements of shoreline public spaces such as distance to water, accessibility, and green coverage ratio [7,30]; and spatial elements of waterfront inland such as population, land functions, and building density [7,25,30,35,44] are also considered. Some studies revealed the influence of these spatial elements on vitality through multiple linear regressions and other statistical algorithms [45]. Waterfront spatial elements that have a positive impact on space vitality include greening rate, pedestrian lane density, proximity to water, various service facilities, population, and the surrounding building density [7,25,30].

However, the aforementioned studies and their conclusions on the relationship between the two tend to involve the surrounding factors of the waterfront and are mostly based on a specific time, such as the waterfront vitality measurements of the Huangpu River in Shanghai [7], Lake Jinji in Suzhou [25], Yangtze River in Nanjing [29], and a waterfront in Wuhan [30]. Thus, intuitive evidence that reveals the influences without a tracked perspective is lacking. This may be due to the difficulty of collecting long-term data to track the construction sequence of the project, which tends to ignore the impact of the spatial elements' changes on vitality. Moreover, unlike other elements within the waterbody and inland surroundings, few quantitative studies focused on shoreline public spaces, which are mainly design contents involving a waterfront renewal process for planners and architects. This might be because the spatial scale of the waterfront open space is mismatched with an insufficient resolution ratio of the current data because of its sparse attributes [26] in terms of the morphological characteristics and public activities, compared with building density, population, and property price in waterfront surroundings [3].

Therefore, our study collects two-time datasets before and after a waterfront renewal to focus on the changes in the involved spatial elements. In addition, attention to the changes can partly remove subjectivity while developing the index framework and assessment rules. Moreover, compared with using the absolute value of single-time-point data, documenting the relative value with a comparison between two situations can, to some extent, offset the inaccuracy caused by the sparse attributes of the spatio-temporal data themselves [26].

## 3. Methodology

Our study included three steps: case selection, index framework development, and data collection and treatment (Figure 1). The Maozhou River waterfront renewal was the subject. We developed a spatial element evaluation framework and space vitality index framework for its renewal. Data related to physical spatial elements and functional activities were collected before and after the renewal and processed with an optimization algorithm. By comparing waterfront space vitality in different years, the revitalization and changes in spatial elements and space vitality were analyzed.

### 3.1. Two-Time Case Selection

The Maozhou River project in Shenzhen, constructed from 2019 to 2020, is an early project of the "Ten-thousand km Garden-Way" program proposed by the Guangdong government [46–49]. By selecting this project—a recent representative renewal case in China, converting black and odorous rivers into a quality public space—our study provides guidance for future waterfront renewal projects. This study focuses on the project's major design of shoreline public spaces, the so-called key research units, 61 in total, which are spread inland 50~10 m from the water border over a 1.3 km² area. Furthermore, to monitor the influenced vitality of the waterfront renewal, we delineated an influenced research area, which was 500~1000 m inland from the water border based on the concept of building a 15 min neighborhood, covering a 15.8 km² area (Figure 2).

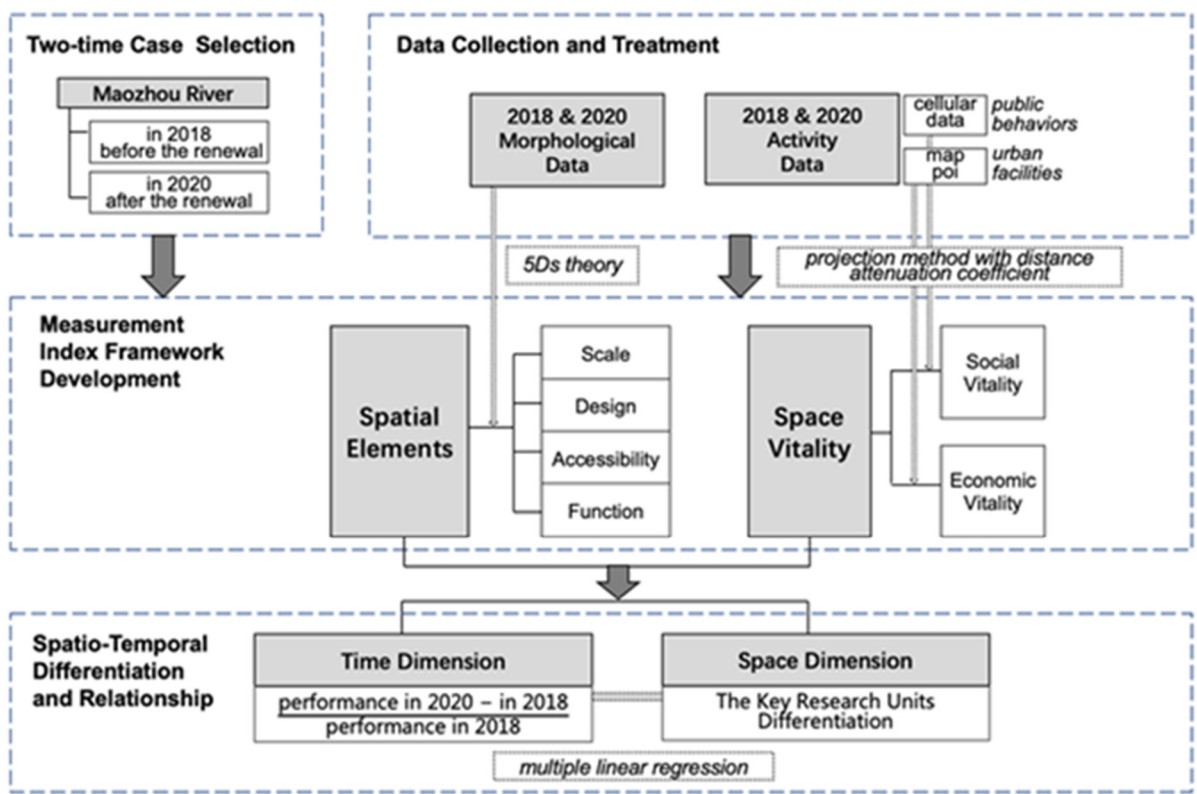

**Figure 1.** Research path and methodology.

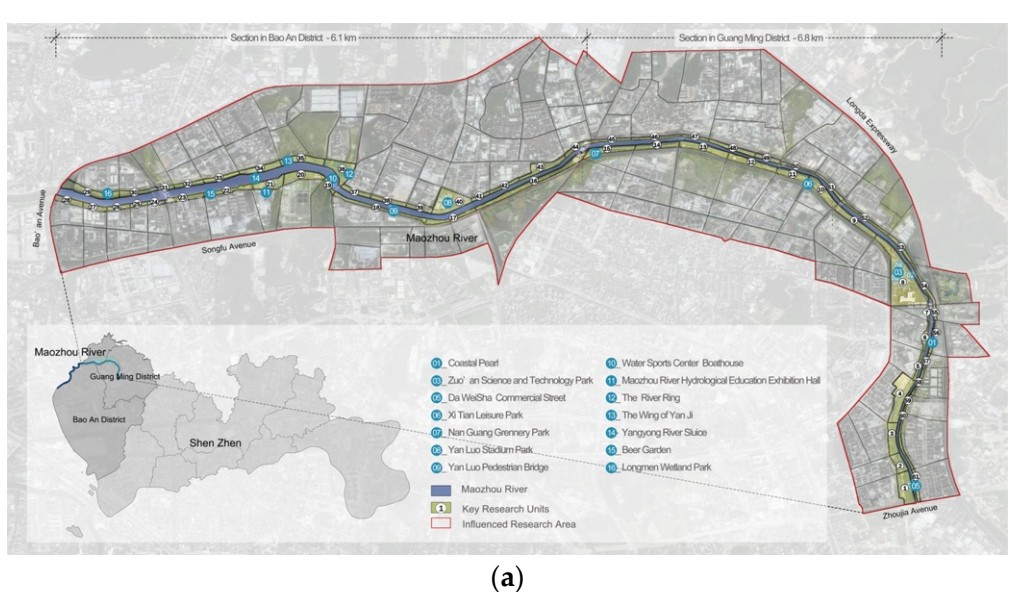

(**a**)

**Figure 2.** *Cont.*

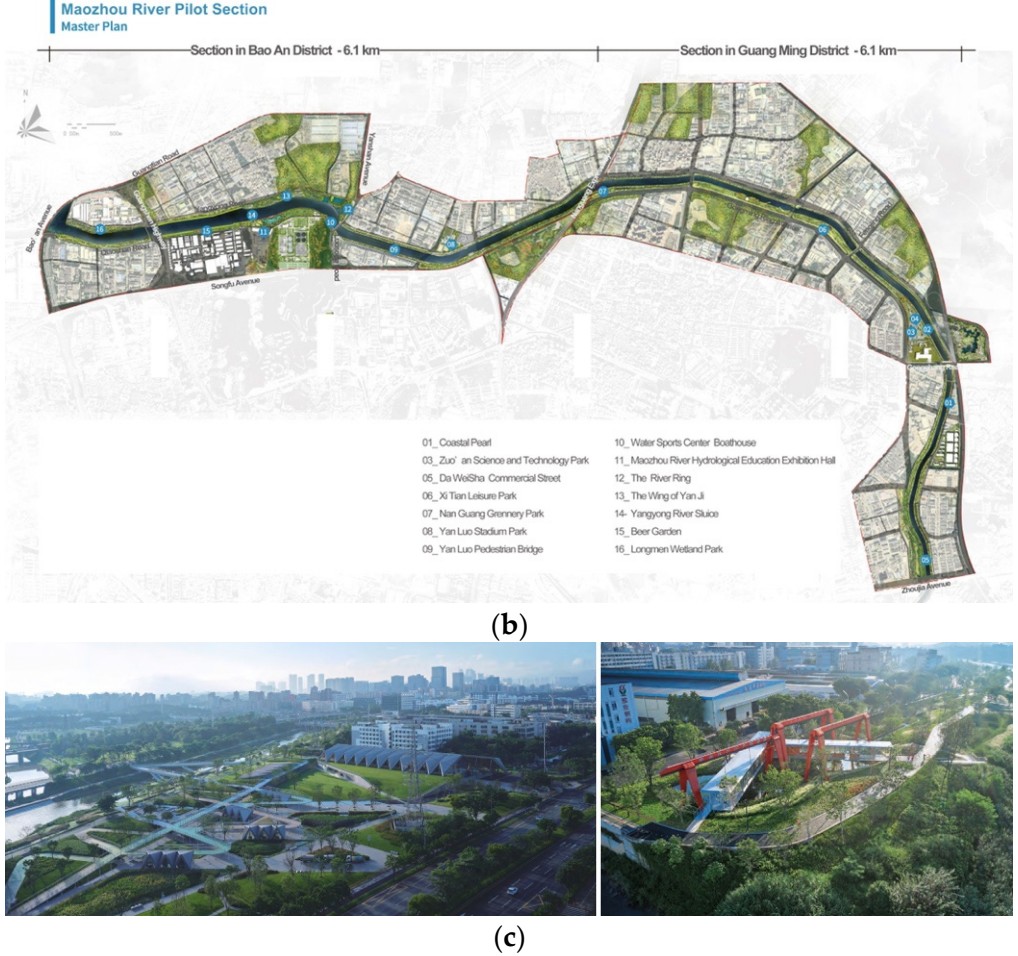

**(b)**

**(c)**

**Figure 2.** Research case information. (**a**) Research case scope, (**b**) The masterplan of the Maozhou River renewal project (cited from Tongji Architectural Design (Group) Co., Ltd.), (**c**) The images of the typical node spaces of the project, Left Bank Park (cited from reference [47]) and Longmen Wetland Park (cited from reference [48]).

*3.2. Measurement Index Framework Development*

3.2.1. Spatial Element Evaluation Framework for the Waterfront Renewal

We selected the shoreline public space as our key research area to develop an evaluation framework because this refers to the actual scope where the spatial elements have been improved during the waterfront renewal. We developed an index framework by adapting the framework of built-environment quality [38] and the high-quality waterfront design principles [5,40,41], and by following the actual design methods for the Maozhou River [47–49]. Waterfront spatial typology and formation were evaluated based on four dimensions, which were further broken down into 11 elements that constituted the spatial element evaluation framework (Table 1). These dimensions are described as follows.

**Table 1.** Spatial element evaluation framework for waterfront renewal.

| Target Layer | Factor Layer | | Formula | Explanation |
|---|---|---|---|---|
| Scale | Shoreline Breadth | $X_1$ | $\overline{D_i} = \sum_{j=1}^{n} D_{ij}/n$ | $\overline{D_i}$ denotes the average shoreline width of unit i; and $D_{ij}$, the shoreline width of an equally distanced segment j in unit i. |
| Design | Proximity to Water | $X_2$ | $W_i = \sum_{j=1}^{n} w_j l_j / l_i$ | $W_i$ denotes the proximity to water of unit i; $w_j$, the proximity to water of segment j in unit i; and $l_j$ and $l_i$, the shoreline length of segments j and i, respectively. |
| | Path Density | $X_3$ | $\rho_{sli} = S_{sli}/S_i$ | $\rho_i$ denotes the density in unit i; $S_{sl\ i}$, the area of paths—walking, running, and bicycle; $S_{eei}$, the number of path entrances and exits; and $S_i$, the area of unit i. |
| | Path Entrance and Exit Density | $X_4$ | $\rho_{eei} = S_{eei}/S_i$ | |
| Accessibility | Traffic Lane Continuity | $X_5$ | $C_i = 1/c_i$ | $C_i$ denotes the traffic lane continuity of unit i; and $c_i$, the number of bridges connecting roadways and paths on both sides. |
| | Subway Station Coverage | $X_6$ | $\rho_{sbwi} = S_{sbwi}/S_i$ | $S_{sbwi}$, $S_{busi}$, $S_{parking\ i}$ denote the number of subway stations, bus stops, and parking lots, respectively, underground or aboveground, in the service buffer zone available for unit i. |
| | Bus Stop Coverage | $X_7$ | $\rho_{busi} = S_{busi}/S_i$ | |
| | Parking Facility Density | $X_8$ | $\rho_{pki} = S_{pki}/S_i$ | |
| Function | Commercial and Leisure Facility Density | $X_9$ | $\rho_{cmci} = S_{cmci}/S_i$ | $S_{cmc\ i}$ denotes the number of commercial and leisure facilities—catering, shopping, accommodation, and sports and leisure; $S_{ct\ i}$, that of cultural and educational facilities— science, education, culture, and landscape; $S_{sv\ i}$, that of public service facilities—newsstands, washrooms, and emergency shelters. |
| | Cultural and Educational Facility Density | $X_{10}$ | $\rho_{cti} = S_{cti}/S_i$ | |
| | Public Service Facility Density | $X_{11}$ | $\rho_{svi} = S_{svi}/S_i$ | |

Notes: Regarding the performance of proximity to water: When the stormwater infrastructure is exposed, the height exceeds 1.5 m, and water is almost unseen in all sections, the hydrophilicity is weak and set as 1. When the stormwater infrastructure is exposed, the height is between 1.1 and 1.5 m, and a broad area of water is visible although sight is blocked in a few sections, the hydrophilicity is average and set as 2. When the stormwater infrastructure is exposed, the height is lower than 1.1 m, and a broad area of water is visible, the hydrophilicity is strong and set as 3. When the stormwater infrastructure is integrated into the landscape design, and a broad area of water is visible and suitable for water activities, the hydrophilicity is set as 4.

(1) Scale: Built-environment scale and density is an important index that influences user behavior [50]. For open, linear public spaces such as waterfront spaces, the width of the shoreline open space from the waterbody to inland (shoreline breath) was assumed to influence space use.

(2) Design: It is suggested that improved environment and public space quality promotes more activity and space utilization [16]. In waterfront renewal projects, close and wide views of the waterbody [51] and numerous connected walking, running, and bicycle paths with many entrances and exits create comfortable environments that promote leisure activities [5,41].

(3) Accessibility: Accessibility is an important factor in built-environment measurements [21,52]. According to the sampled space characteristics, we measured accessibility using key indexes: the number of bridges and paths that connect the two river banks, the effective coverage of public transportation such as subway and bus stations in the buffer zone, and the number of buffer zone parking facilities [7,31]. The service buffer zone of the subway stations, bus stops, and parking lots is set according to the standard of the daily walking distance, i.e., 800 m, which is a suitable distance for a 10–15 min walking range.

(4) Function: Land function diversity is an important factor for urban vitality [18,35]. We assumed that even for a public waterfront space with only one land-use purpose, functionally diverse design would be effectively the same. Accordingly, functional facilities related to commerce, entertainment, culture, education, and public services were incorporated into indexes to represent waterfront space renewal.

### 3.2.2. Space Vitality Evaluation Framework for Waterfront Renewal

Waterfront space vitality is defined by the levels of public activities and urban functions, namely, social vitality and economic vitality [3,16,18]. As for social vitality, user activities in a space can be divided into "pass-by" and "stay". Instead of "pass-by", the level of "stay" activities in a space is often used to evaluate space quality [12]. Therefore,

here, the population that engages in "stay" activities, such as dwellers, workers, or visitors, is used to represent social vitality intensity, and the total length of time visiting users engage in "stay" activities is used to represent social vitality quality. For economic vitality, enterprise sites, commercial services, culture and education, leisure, and public services can reflect the activity level of urban functions [31,45]. Hence, the number of facilities is used to evaluate economic activity intensity, whereas the functional diversity, as measured by Shannon entropy based on mixed-use land calculations in urban research, is used to represent economic activity quality [34,35] (Table 2).

**Table 2.** Space vitality evaluation framework for waterfront renewal.

| Target Layer | | | Factor Layer | | Data Source | Explanation |
|---|---|---|---|---|---|---|
| Social Vitality | Vitality Intensity | $Y_1$ | Number of Dwelling Users' Stay | $Y_{11}$ | Cellular data | The population of users who dwell in, work at, or visit the area |
| | | | Number of Working Users' Stay | $Y_{12}$ | | |
| | | | Number of Visiting Users' Stay | $Y_{13}$ | | |
| | Vitality Quality | $Y_2$ | Length of Visiting Users' Stays during Weekdays (Day) | $Y_{21}$ | | The length of visitors' stay |
| | | | Length of Visiting Users' Stays during Weekdays (Evening) | $Y_{22}$ | | |
| | | | Length of Visiting Users' Stays during Weekends (Day) | $Y_{23}$ | | |
| | | | Length of Visiting Users' Stays During Weekends (Evening) | $Y_{24}$ | | |
| Economic Vitality | Vitality Intensity | $Y_3$ | Number of Enterprises | $Y_{31}$ | Map POIs | POIs' number—companies or enterprises |
| | | | Number of Commerce Facilities | $Y_{32}$ | | POIs' number—catering, shopping, accommodation, sports, and leisure facilities |
| | | | Number of Culture and Education and Recreation Facilities | $Y_{33}$ | | POIs number—for culture, education, and recreation |
| | | | Number of Public Service Facilities | $Y_{34}$ | | POIs number—public facilities, life services, governmental organizations, and social communities |
| | Vitality Quality | $Y_4$ | Diversity of Economic Function Facilities | $Y_{41}$ | Equation: $m_i = \dfrac{-\sum_{j=1}^{k} q_j \ln(q_j)}{\ln(k)}$ | $m_i$ denotes the functional diversity in the influence area of unit i; and $q_j$, the ratio of the number of facilities of the $j^{th}$ function type to the number of the total facilities in the influence area of unit i. |

Notes: Regarding the population of users: In principle, if users stay at the same location for more than 30 min, it is identified as a "stay" and added to the stay length of the user of the cellular tower. According to the monthly stay length, the "stay" type is categorized as "dwelling", "working", and "visiting". If a user stays at a place from 9:00 pm to 8:00 am the next day (9:00 am to 5:00 pm), the "stay" is marked as "dwelling" ("working"). Others are marked as "visiting".

### 3.3. Research Data Collection and Treatment

#### 3.3.1. Data Collection

Based on the spatial element and space vitality frameworks, we collected the same categories of morphological and activity data for corresponding periods in 2018 and 2020. Morphological data were collected from remote sensing images, which were combined with on-site survey maps and project design documents. Data from public sites, such as the Shenzhen Statistic Bureau website, were used as supplements. Thus, we obtained morphological data (i.e., shape, size, and function) of basic sites including road networks and land boundaries as well as major—planned or built—public spaces. For activity data, we collected two kinds of data, respectively, in October 2018 and October 2020, which are cellular data from a selected communication operator and POIs on maps. We used monthly spatio-temporal cellular data, statistically overlaid with the instantaneous data from each cell tower, which effectively helps avoid some daily accidental errors. It is worth noting that between 2018 and 2020, during the waterfront regeneration, 24 new cell towers

had been installed. After manually deleting these towers, 443 effective cell phone signal towers in the influenced research area remained, each with a precision range of 200 to 500 m. In total, 829,539 and 766,388 pieces of cellular data were collected in 2018 and 2020, respectively, including anonymous user IDs of encrypted mobile phones, timestamps, cell tower geographical coordinates, stay length, and stay type. In collecting POI data, we first applied web service access keys and then obtained POI data from maps in October 2018 and October 2020 using Python. Overall, 10,772 and 10,244 pieces of POI data of the influenced research area were selected in 2018 and 2020, respectively (Figure 3).

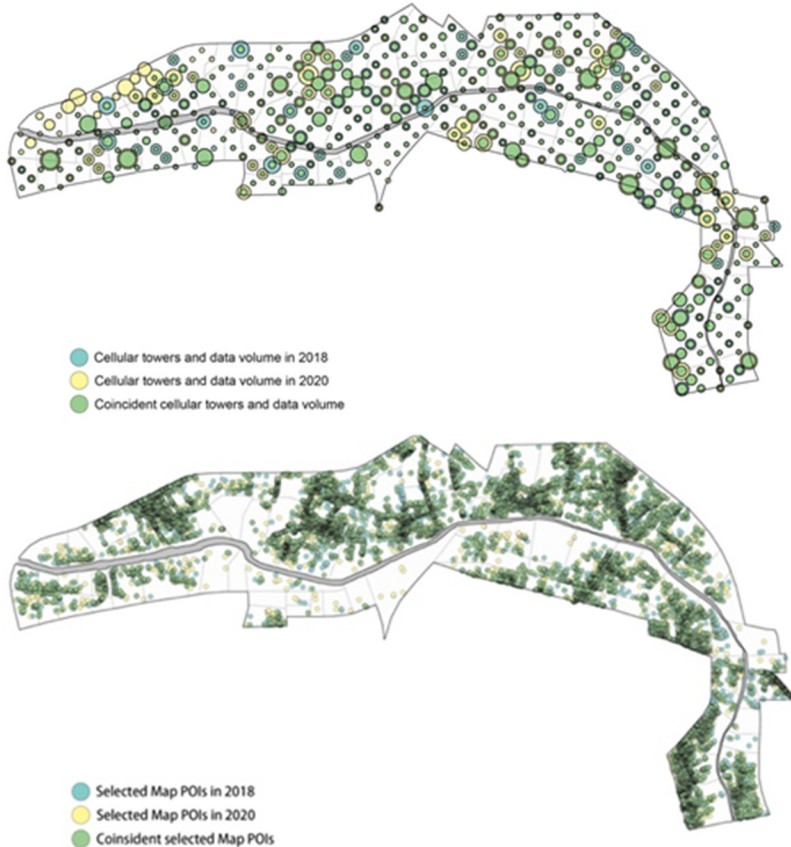

**Figure 3.** Cellular data and map POI data.

### 3.3.2. Activity Data Processing: Influenced Research Area Projection Method with Distance Attenuation Coefficient

Note that the waterfront vitality measured here is the vitality evaluation of a certain area (the influenced research area) brought by the improvement of waterfront public space renewal (key research units). Thus, the next step is to convert the activity data collected in the influenced research area into each key research unit.

From previous research on waterfront public space and waterfront inland, we found that the domain is generally divided into 2 or 3 research belts by a pedestrian shed of 5, 10, and 15 min [7,17]. Because this stepped segmentation method assigns the same vitality and space characteristics to a belt, it disregards the revival of regional vitality brought by public space renewal, which changes continuously with distance. Instead of simply combing the activity data within the pedestrian shed, we considered the distance attenuation when converting the data from the influenced research area to the key unit. A gravity model was initially proposed to reflect the retail gravitation on its neighborhood [53], interpreting that the potential attraction between spaces was negatively related to distance. Gradually, this has been widely recognized and applied in urban geography studies and urban planning such as infrastructure accessibility [54] and origin–destination flows [55]. In these studies, the law of distance attenuation is constantly confirmed, and the attenuation curve

is gradually fitted with the power function. Therefore, considering the regional vitality impact of the waterfront renewal, we introduced this distance attenuation coefficient from the gravity model, so that the impact represented by the activity data can decrease exponentially with distance in the influenced research area (Figure 4). The equations are as follows:

$$A_i = \sum_{j=1}^{n} a_{ij} \tag{1}$$

$$a_{ij} = V_j / d_{ij}{}^{\beta} \tag{2}$$

where $A_i$ denotes the value of the $i^{th}$ unit after the influence research area projection; $a_{ij}$ is the projected value of the $j^{th}$ data point on the $i^{th}$ unit; $V_j$ is the original value of the $j^{th}$ data point; $d_{ij}$ is the distance of the $j^{th}$ to the $i^{th}$ unit; and $\beta$ is the distance attenuation coefficient. $\beta$ varies depending on the application scenarios, but many practices found that if $\beta$ ranged from 1.5 to 2, it had little effect on the results [56]. Therefore, here, we generally set $\beta$ as 1.8.

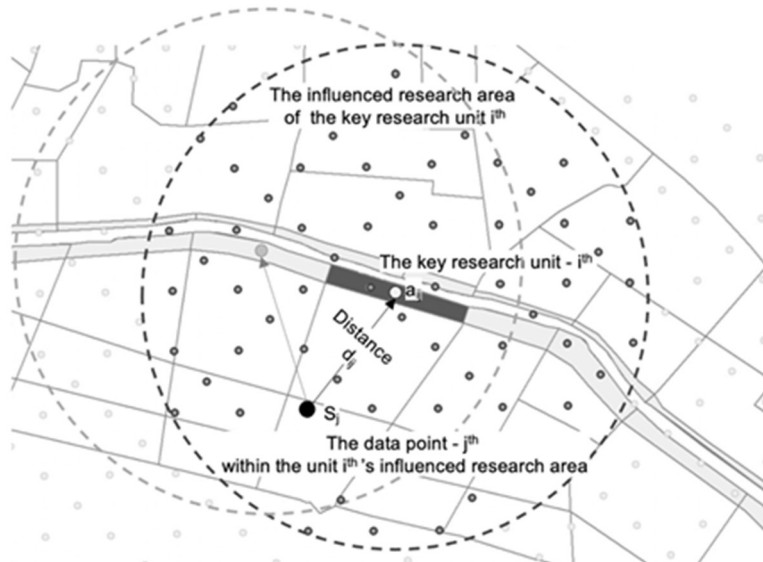

**Figure 4.** Influenced research area projection method with distance attenuation coefficient.

3.3.3. Temporal Comparison Analysis and Relationship Model

For the temporal comparison analysis of spatial elements and space vitality before and after the renewal of waterfront space, we used the year-on-year growth rate:

$$\Delta_i = (A_{2020i} - A_{2018i}) / A_{2018i} * 100\% \tag{3}$$

where $\Delta_i$ denotes the increased rate of the $i^{th}$ factor, and $A_{2020i}$ and $A_{2018i}$ denote the performance of the $i^{th}$ factor in 2018 and 2020, respectively.

We used the most popular multiple linear regression to reveal the linear relevance between two sets of variables [45]. First, the relationship model was built with the increased rate of waterfront renewal spatial elements as independent variables and the increased rate of space vitality as dependent variables. Second, multicollinearity and Durbin–Watson tests were used to remove correlated independent variables. Finally, critical influential elements were obtained from multiple linear regression models.

## 4. Results and Analysis

### 4.1. The Results of Space Vitality Changes

The four space vitality categories increased significantly from 2018 (before the renewal) to 2020 (after the renewal; Figure 5). Social vitality intensity and quality, and economic vitality intensity and quality increased by 28.09%, 16.98%, 27.24%, and 3.53%, respectively.

As suggested by the change in the "society vitality" and "economic vitality" relationship (Figure 6a), the research units' values were generally distributed alongside the x-axis, with a few diverging toward the y-axis, suggesting that social vitality changed more than economic vitality. Although the economic activities in most units became increasingly diverse, the change was gradual, and no large-scale growth appeared. As shown by the change in the "vitality intensity" and "vitality quality" relationship (Figure 6b), more data fell into the first and third quadrants. Data in the first quadrant significantly diverged, indicating that the vitality of some units surged. Regarding the stay length and population of different user types at different times (Figure 7a,b), visitors were more prevalent than dwellers or workers, and visitors stayed much longer in the daytime and weekend evenings than on workdays. Notably, economic vitality, as represented by the number of POIs of the four categories, dropped from 2018 to 2020 (Figure 7c). The number of enterprises and commercial services decreased but the number of cultural, educational, and recreational facilities (exhibition venues and parks) increased by 123.41%, on average.

Generally, the space vitality of the overall waterfront has improved significantly, even with the impact of the COVID-19 pandemic. After renewal, the space attracted more visitors, especially during weekends, and garnered longer "stay" times. The number of economic functions and activities of most units somewhat decreased in the two years, and activity diversity increased for most units, albeit to a limited extent. The influence of the broad environment, including the COVID-19 pandemic, can explain the large-scale decrease in enterprise numbers; however, function replacement in some node places can likewise offer other important explanations.

Meanwhile, the results also show rapid changes in some node spaces, which are consistent with the base conditions and the space design vision. With substantial effort and cost put into the design and construction, new projects such as the River Ring [49] generated abrupt overall vitality promotion. Such promotion occurred despite the significant decrease in economic vitality intensity that happened in the Longmen Wetland Park [48] and Left Bank Science and Technology Exhibition Hall [47], where industrial wharves and warehouses were originally regenerated into exhibition halls and parks were open to the public.

### 4.2. Regression Model Results

To reveal more precisely the influences of spatial element renewal on the social and economic vitality of public spaces, we set up 15 regression models for spatial elements, relevant to user cellular data in the key research area processed based on the Thiessen polygons algorithm projection and POI data in the influenced research area, according to the distance attenuation projection (Table 3). From the Durbin–Watson test results, all models' D–W values were close to 2, suggesting no collinearity or autocorrelation between these independent variables. The *p* values for the significance test indicated the validity of all other models except the one with $Y_{31}$ as the dependent variable.

**Table 3.** Regression models' test results and relevance degree.

| | 1 | 2 | 3 | 4 | 5 | 6 | 7 | 8 | 9 | 10 | 11 | 12 | 13 | 14 | 15 |
|---|---|---|---|---|---|---|---|---|---|---|---|---|---|---|---|
| | Y11 | Y12 | Y13 | Y1 | Y21 | Y22 | Y23 | Y24 | Y2 | Y31 | Y32 | Y33 | Y34 | Y3 | Y4 |
| D-W Value | 1.72 | 1.64 | 1.37 | 1.64 | 1.65 | 1.67 | 1.58 | 1.56 | 1.63 | 2.18 | 1.65 | 1.81 | 1.66 | 1.65 | 1.39 |
| *p* Value | 0.000 b | 0.000 b | 0.000 b | 0.000 b | 0.000 b | 0.000 b | 0.000 b | 0.000 b | 0.000 b | 0.144 b | 0.000 b | 0.002 b | 0.000 b | 0.000 b | 0.000 b |
| R-squared | 0.51 | 0.61 | 0.70 | 0.60 | 0.62 | 0.64 | 0.69 | 0.70 | 0.65 | 0.26 | 0.70 | 0.43 | 0.84 | 0.81 | 0.52 |
| Adjusted R-squared | 0.41 | 0.52 | 0.63 | 0.51 | 0.54 | 0.55 | 0.63 | 0.63 | 0.57 | 0.09 | 0.63 | 0.30 | 0.81 | 0.76 | 0.41 |

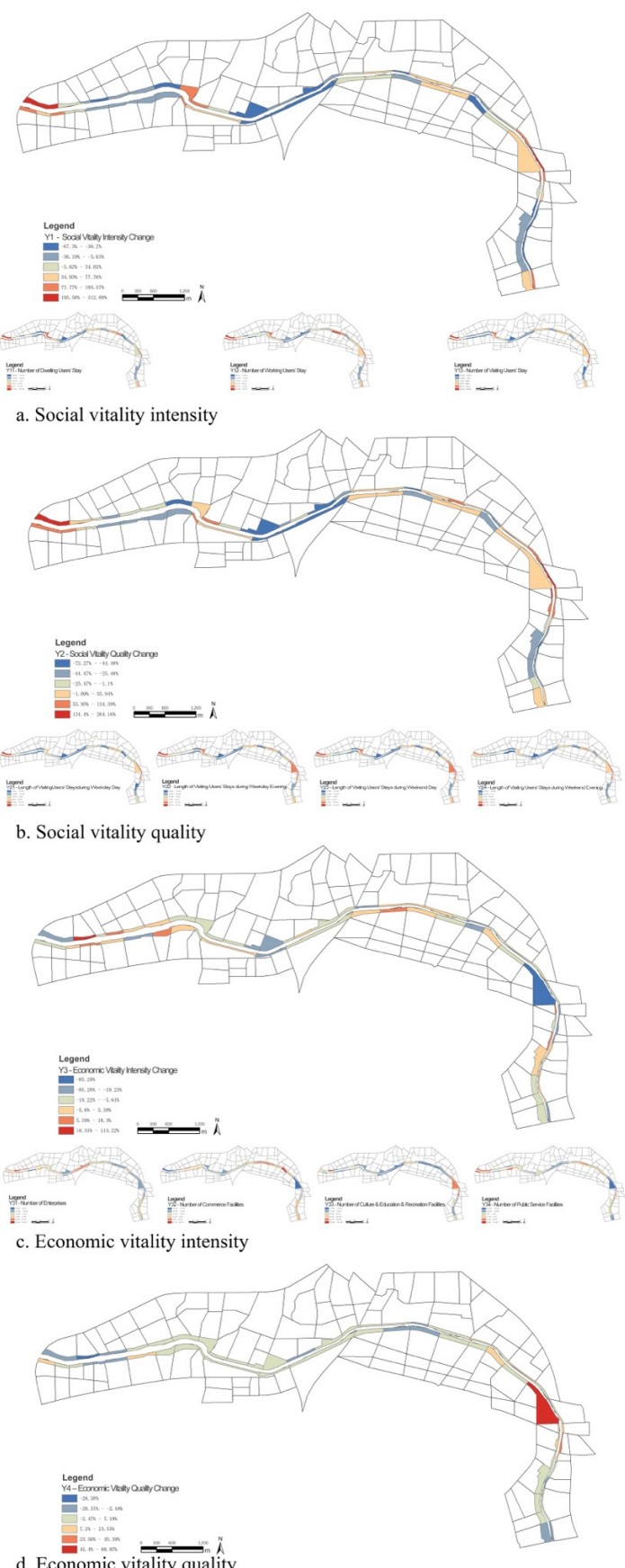

**Figure 5.** Change in Maozhou River waterfront space's vitality from 2018 to 2020.

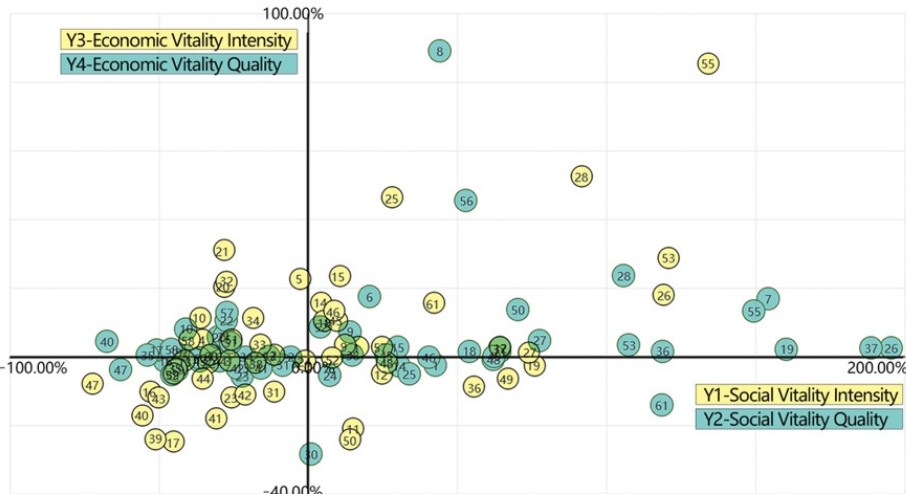

a. Relationship between "social vitality" and "economic vitality"

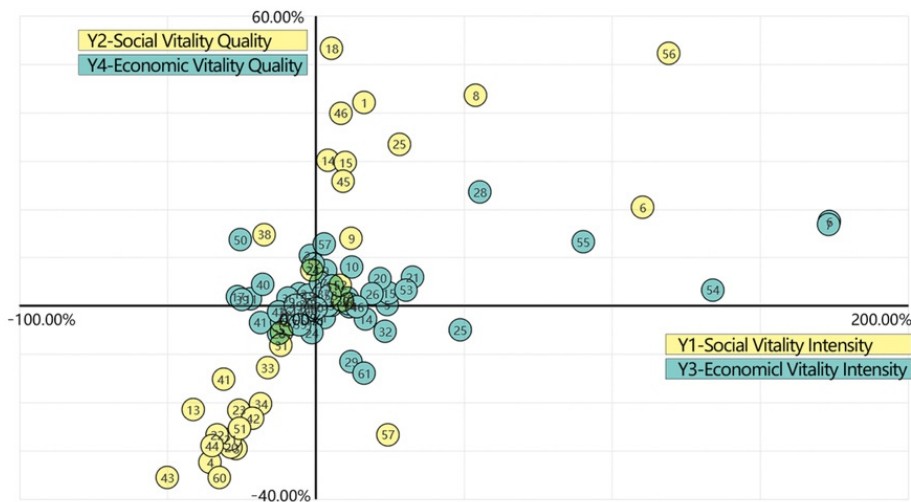

b. Relationship between "vitality intensity" and "vitality quality"

**Figure 6.** Quadrant diagram of the changes in space vitality.

As suggested by variations in adjusted R-squared, the strength of the relationship varied. Spatial element renewal better explained the improvement of visitor activity quality during weekend days and evenings than on weekdays. There was greater relevance of the spatial element to visitor activities compared with dwellers or workers. The impact of spatial elements through the improvement of economic vitality intensity was clearer than that through economic activity diversity. When economic activities were further categorized, the relationship between space renewal and public services was significantly high.

Moreover, for each spatial element in the 15 models, no multicollinearity with a variance inflation factor higher than 6 was found. Moreover, all spatial elements in a certain model displayed some correlations with *p* values less than 0.1 or even 0.05. Specifically, for the nine models on social vitality, $X_5$, $X_8$, and $X_{11}$ barely showed an influence relationship with *p* values more than 0.1, and for the other six models on economic vitality, neither did $X_2$, $X_4$, and $X_7$.

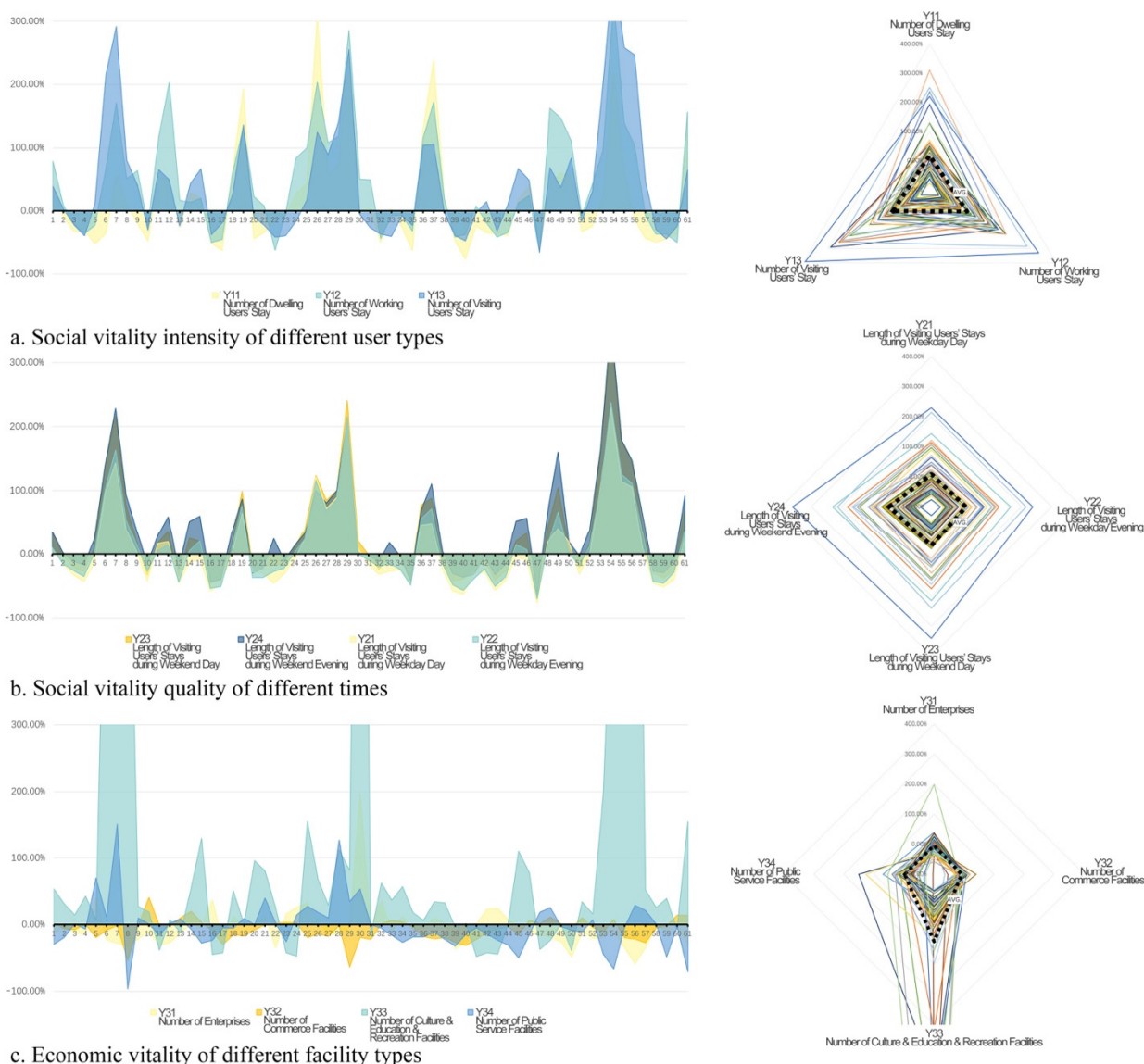

**Figure 7.** Space vitality of different factor types.

## 5. Discussion

### 5.1. Relationship between Space Vitality and the Spatial Elements

Despite the COVID-19 pandemic's effects, the waterfront space vitality results suggest a general increase of more than 20% as a whole, and rapid changes in various vitality factors and in some node spaces after the completion of the Maozhou River renewal project, which are consistent with the base conditions and the space design vision. Moreover, the spatial elements' improvement during renewal construction varied in their extent of influence on the space vitality. First, the influence of spatial element renewal on economic vitality intensity was opposite to its influence on the other three space vitality factors (Figure 8a). Second, in terms of the influence on social vitality intensity by user type, the spatial elements' changes had a more obvious influence on dwellers than on visitors or workers (Figure 8b). Moreover, the influence on social vitality quality by users' stay during different times showed similar characteristics (Figure 8c). Finally, among the influences on economic vitality intensity by functional facility, the differences were substantial, especially for culture, education, and recreation facilities (Figure 8d).

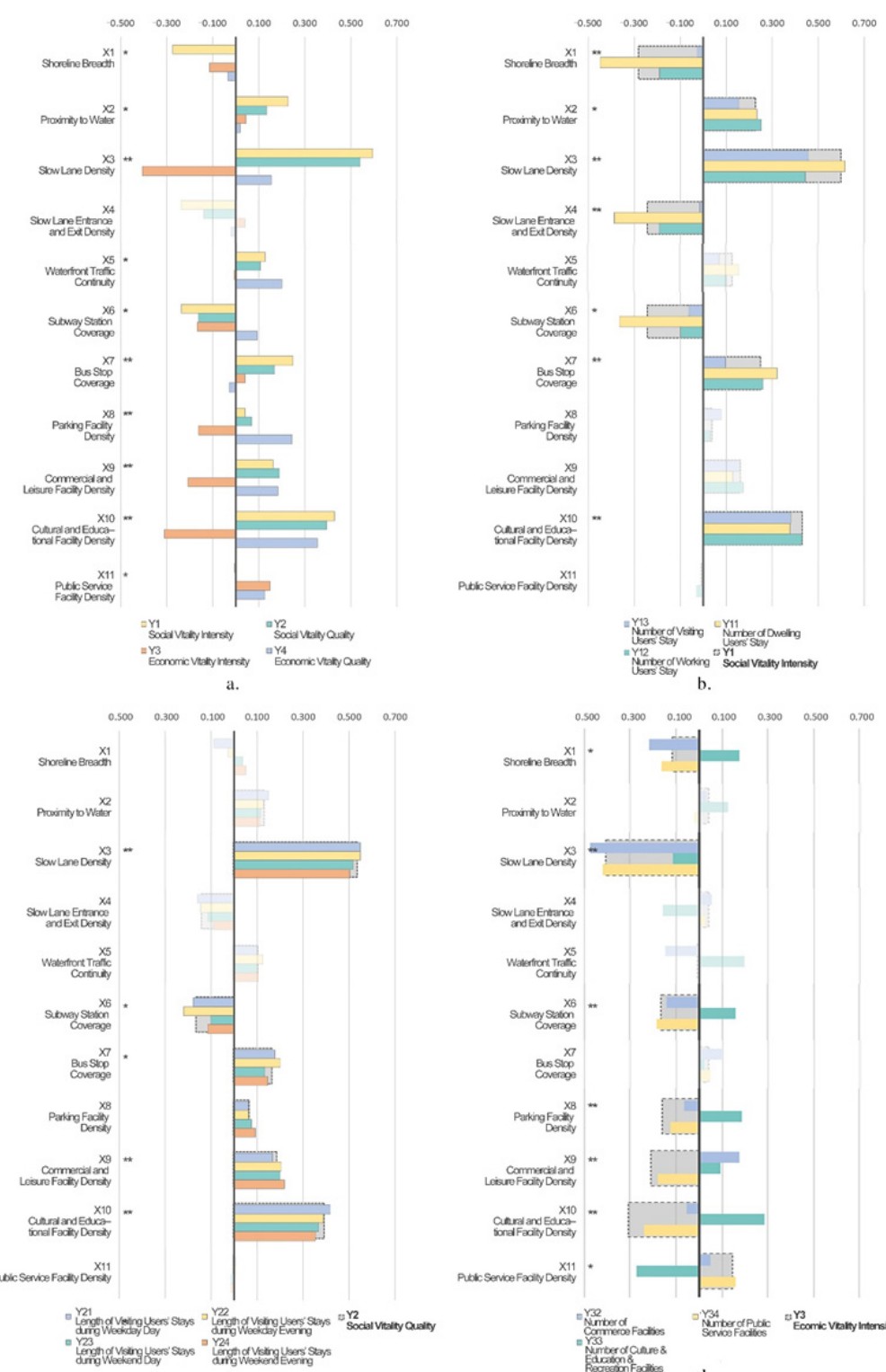

Notes: ** and * denote $p < 0.05$ and $p < 0.1$, respectively. Figures a, b, c, and d above illustrate the influence of spatial elements on four space vitality categories, on social vitality intensity of different user types, on social vitality quality of different times, and on economic vitality intensity of different facilities, respectively.

**Figure 8.** Influence of spatial elements on space vitality.

In general, based on these different space vitality changes, the improvement of spatial elements during renewal construction showed some common features of the trend of the

influences, in which performance was broadly "Function" > "Design" > "Accessibility" > "Scale" (Figure 8). In the "Function" dimension, the "commercial and leisure facility density" and "cultural and educational facility density" had the most significant positive influence on space vitality. Social vitality was subject to the relatively significant positive influence of some factors from the "Design" dimension, such as "path density" including walkways, runways, and bicycle ways, and "proximity to water" representing the field of view and closeness to the waterbody, which is also a typical and special factor for waterfront various from other public spaces. In terms of "Accessibility", though the "bus stop coverage" within the pedestrian buffer zone has a relatively positive influence, spatial elements that had weak or even suppressive influences were "path entrance and exit density" and "subway station coverage".

*5.2. Findings on the Elements' Influence Compared with Prior Research*

(1) The "Function" dimension elements' improvement has the most significant influence on space vitality in the Maozhou River waterfront. Many waterfront areas are narrow, elongated, and monotonous, and lack the necessary public services and activity spaces. For example, the Dasha River waterfront in the center of the Nanshan district of Shenzhen has been overwhelmed with driveways and has become so narrow that it can only be used as a passageway and is not a place where people can stay, which is not the same situation as those discussed in other case studies [57]. In the Maozhou River waterfront, large venue nodes were built every 500–800 m and smaller ones every 200–300 m, including public toilets, coffee shops, kid entertainment spaces, sports and leisure facilities, exhibition halls, and science education facilities. We find that these functional places serve as "catalysts" that stimulate a surge in staying behavior and activities, which is similar to findings in other research [7,10].

(2) The "Design" dimension elements, such as path density and shoreline proximity to water, have positive influences, while the "Scale" dimension is less effective or even suppresses vitality. If "proximity to water" is calculated based on the field of view and accessibility, its influence is significant, as same as the analysis in the prior research [7], but if based on shoreline distance ("shoreline breadth" here), the relationship is weak, which is the same result but in a quite different dimension compared with the prior research [25]. Shoreline breadth represents the size of the linear public space on the waterfront. If the space is too large (small), it can reduce accessibility and proximity to water (be difficult to realize design and diversity). Shoreline breadth is often an important indicator in urban design or even in urban regulation and control and is worthy of further discussion, as it plays a determining role in the design and positioning of waterfront spaces.

(3) The "Accessibility" dimension elements have slightly different effects. Bus stop and parking facility coverage have significant impacts while subway station coverage seems useless, thus contradicting the findings of current public space (streets and waterfront) research in downtown areas from other studies [24,28]. The Maozhou River is in a Shenzhen suburb, where subway station density is quite low compared with other areas in the city. Almost all subway stations in the research area are unsuitable for walking at a 500 m distance from public spaces and thus fail to increase the vitality of the central research area. Our findings offer a new influence mechanism of waterfront renewal in the suburbs, especially when the focus of existing space renewal gradually moves from the downtown to the suburbs and new towns.

## 6. Conclusions and Limitations

In general, our work provides a relatively creative method as a kind of post-occupancy evaluation helping find the most useful spatial elements for guiding the next waterfront space planning and design. This method is the use of before and after (two-time) datasets to measure space vitality and the influenced relationship of spatial elements from a spatiotemporal differentiation perspective, which can be also extended to evaluate other linear public space renewal projects. Moreover, because China's urban renewal program has

been spreading from central cities to the suburbs, the Maozhou River case is representative of waterfront renewal in the suburbs, our findings (e.g., to increase functional diversity and improve space design) can guide new waterfront renewal initiatives, especially in the suburbs.

The Maozhou River project, which converted black and odorous rivers into a quality public space, is a representative renewal case, offering insights for improving space vitality research from a spatio-temporal differentiation perspective. However, this study was limited by time and type of data collection. Specifically, the massive effect of the COVID-19 pandemic on vitality could not be removed. In future studies, the study areas' gross domestic product or other overall economic indicators can be used as fundamental bases for further analysis. Moreover, we are planning to continuously monitor this project for the next 5 or 10 years to determine any changes in the results upon the improvement of the area's vitality from increased business and cultural activities and higher diversity, especially to rule out the impact of the pandemic. Furthermore, texts, images, and videos on social media can be used as a data source for evaluating users' subjective feelings and comments from a humanistic perspective to enhance space renewal measurement. In addition, the two frameworks can be improved. Aquatic ecology elements such as water quality and fauna and flora diversity [28,43] should be added, as they are major considerations in the environmental construction and evaluation of ecological improvement and should, therefore, should be used as spatial elements and measures of benefits in future public space renewal design. Finally, the optimization algorithm for cellular signal data and others presents a difficult problem; although we adopted the innovative distance attenuation projection approach from the gravity model, further testing and verification are needed. For further research, we still need to extend our thinking and methodology to combine qualitative research with quantitative measurement, which is a highly complex challenge and involves many learning contributions in different levels of critical thinking from local users' or researchers' knowledge.

**Author Contributions:** Conceptualization, resources, data download and curation, visualization, and writing—original draft preparation, Y.F.; methodology and software, Y.F., W.T. and Y.Y.; writing—review and editing, Y.F., D.K. and Y.Y; supervision, D.K. and Y.Y. All authors have read and agreed to the published version of the manuscript.

**Funding:** This research was funded by National Natural Science Foundation of China [52078343], Fundamental Research Funds for the Central Universities [22120210540], National Natural Science Foundation of Guangdong [2023A1515011274], and Graduate Education Innovation Program of Guangdong [2021SFKC072].

**Data Availability Statement:** Data is unavailable due to privacy restrictions. If you are interested, please contact the corresponding author.

**Conflicts of Interest:** The authors declare no conflict of interest.

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
