# Peer review of "Which Spatial Elements Influence Waterfront Space Vitality the Most?—A Comparative Tracking Study of the Maozhou River Renewal Project in Shenzhen, China"

_land, doi:10.3390/land12061260_

Round 1

Reviewer 1 Report

Working in a scenary of scientific discussion, the article WICH SPATIAL ELEMENTS INFLUENCE WATERFRONT REVITALIZATION THE MOST? is highly recommended to publication for its overall merit. The work has strong relevance and significance to readers for its deep theoretical basis, methods description and results presentation.   But, it is still necessary to highlight few points to review.    Regarding to title, it guides readers to expect a deeper contextual reference about "waterfront revitalization" which is a huge concept not exactly developed on the paper. Instead, seems that "space vitality" is much more important to apper in title since it is the main importan evaluation parameter. It will lead readers, since the beggining, to what is really important to consider in spatial elements.   The text is very well done, deep but not very long. But it contains excessive references that sometimes disturbs comprehension. We may suggest reducing to main important ones or, at least, to synthesize them in a conceptual framework. It is also important to highlight that this article brings a very important contribution to disseminate quantitative measurement methods using computer algorithms database, searching for better accurancy. Despite this scientific soundness, researchers must be aware to not underestimate or even downgrade (is not exactly the case here) qualitative methods and subjectivity in science.  To further reflections about it, in multidisciplinary fields such as architecture and urban design, geography, urban planning or territorial studies, each interpretation of any database, even acquired by quantitative methods, is partially subjective, since it is highly dependent of reader's skills and its own context. Moreover, qualitative research is a highly complex challenge and involves many learning contributions in different level of critical thinking from local users or researchers knowledge to empirical, scientifical or philosofical dimensions. In soft sciences, one of the main questions nowadays is "how to reach and what kind of precision is requested from qualitative research?".

Author Response

Reviewer #1:

Working in a scenary of scientific discussion, the article WICH SPATIAL ELEMENTS INFLUENCE WATERFRONT REVITALIZATION THE MOST? is highly recommended to publication for its overall merit. The work has strong relevance and significance to readers for its deep theoretical basis, methods description and results presentation.  But, it is still necessary to highlight few points to review. 

Reviewer #1 comment 1:

Regarding to title, it guides readers to expect a deeper contextual reference about "waterfront revitalization" which is a huge concept not exactly developed on the paper. Instead, seems that "space vitality" is much more important to apper in title since it is the main importan evaluation parameter. It will lead readers, since the beggining, to what is really important to consider in spatial elements.

Response:

Many thanks for pointing out this key issue. It is worth noting that “revitalization” here is to emphasize the change of vitality, which indicates the key point of our creative method to collect the two-time datasets comparatively tracking before and after the renewal construction. But it does confuse readers without exactly developing the paper. As a response, we have replaced “revitalization” with “space vitality”, which is exactly the keywords throughout the paper. Moreover, we also replace “spatio-temporal analysis” with “comparative tracking”, which helps emphasize our creative method more clearly and directly. The new title is attached below:

Which Spatial Elements Influence Waterfront Space Vitality the Most?—A Comparative Tracking Study of the Maozhou River Renewal Project in Shenzhen, China

Reviewer #1 comment 2:

The text is very well done, deep but not very long. But it contains excessive references that sometimes disturbs comprehension. We may suggest reducing to main important ones or, at least, to synthesize them in a conceptual framework. 

Response:

Thank you for pointing out this. We have carefully addressed this question in our response to the editor’s suggestion and you may find more details above. After the two perspectives of revision, reducing some references, and modifying the format, we hope it could be clearer for reading.

Reviewer #1 comment 3:

It is also important to highlight that this article brings a very important contribution to disseminate quantitative measurement methods using computer algorithms database, searching for better accurancy. Despite this scientific soundness, researchers must be aware to not underestimate or even downgrade (is not exactly the case here) qualitative methods and subjectivity in science. To further reflections about it, in multidisciplinary fields such as architecture and urban design, geography, urban planning or territorial studies, each interpretation of any database, even acquired by quantitative methods, is partially subjective, since it is highly dependent of reader's skills and its own context. Moreover, qualitative research is a highly complex challenge and involves many learning contributions in different level of critical thinking from local users or researchers knowledge to empirical, scientifical or philosofical dimensions. In soft sciences, one of the main questions nowadays is "how to reach and what kind of precision is requested from qualitative research?".

Response:

Thank you for pointing out this. It helps a lot to extend our thinking for further work. We do need to consider how to get a precise approach to the combination the qualitative research with quantitative measurement. Thus, we add the text for our future prospect at the end of section 6, line413-416, as follows:

Besides, for further research, we still need to extend our thinking and methodology to combine qualitative research with quantitative measurement, which is a highly complex challenge and involves many learning contributions in different levels of critical thinking from local users or researchers knowledge.

Reviewer 2 Report

Editing problems:

Table 1 - Unable to read the last column well

Fig 7 - Unable to read the graphics (insert graphics not images)

Fig 9 - Unable to read the writed part (insert tables not images)

Comments and suggestions:

It might be interesting and useful to include images of the river regeneration project being considered (project mapping, strategic diagrams, drawings...).

Deepen how the evaluation method identified and described in the article can become a project method.

Then explain, also in the conclusion, the relationship between assessment method and urban planning.

General reflections:

In my opinion two years to verify the success of a major regeneration revitalization project like that are not a fully sufficient time. These projects require medium to long times to produce a truly consolidated and stratified effect in the uses and ways of living a place by the population.

Covid period, as only partially stated in the article, may in my opinion have "distorted" some data and results.

Author Response

Reviewer #2:

Editing problems:

Table 1 - Unable to read the last column well

Fig 7 - Unable to read the graphics (insert graphics not images)

Fig 9 - Unable to read the writed part (insert tables not images)

comment 1:

It might be interesting and useful to include images of the river regeneration project being considered (project mapping, strategic diagrams, drawings...).

Response:

Thank you for pointing out this. Firstly, We apologize for it, we may try our best to cooperate with editors to modify the editing problems. Then, We add three images more together with Fig.2, line 141, one is indicating the overall planning and the other two are the typical node spaces, Zuo’an Park and Longmen Wetland Park. 

Reviewer #2 comment 2:

Deepen how the evaluation method identified and described in the article can become a project method.  Then explain, also in the conclusion, the relationship between assessment method and urban planning.

Response:

Thank you for pointing out this. We may explain this from two perspectives. On one hand, in this article, it is the findings of the evaluation that could become the project method directly. The evaluation method is to investigate the space vitality (i.e. the aim and results of the project) and spatial elements (i.e. project method, the content of planning and design), helping find the most useful spatial elements for guiding the next waterfront space planning and design. On the other hand, the evaluation method provides the potential to develop a post-occupancy evaluation(POE) system, which could become a kind of long-term project method, helping post-evaluate the project and then aiming to improve it next term. So, the following sentences are added and revised at the beginning of section 6, line384-389:

In general, our work provides a relatively creative method as a kind of post-occupancy evaluation helping find the most useful spatial elements for guiding the next waterfront space planning and design. This method is the use of before and after (two-time) datasets to measure space vitality and the influenced relationship of spatial elements from a spatio-temporal differentiation perspective, which can be also extended to evaluate other linear public space renewal projects.

Reviewer #2 general reflections:

In my opinion two years to verify the success of a major regeneration revitalization project like that are not a fully sufficient time. These projects require medium to long times to produce a truly consolidated and stratified effect in the uses and ways of living a place by the population.

Covid period, as only partially stated in the article, may in my opinion have "distorted" some data and results.

Response:

Thanks a lot for your helpful reflection. We very much agree on this and do have plans to continue to track the usage progress of this project into the next 5 or 10 years, especially to rule out the impact of the pandemic. The following sentences in the text stating this deficiency and prospect have been mortified more clearly in section 6, line 396-403, as follows:

However, this study was limited by time and type of data collection. Specifically, the massive effect of the COVID-19 pandemic on vitality could not be removed. In future studies, the study areas’ gross domestic product or other overall economic indicators can be used as fundamental bases for further analysis. Moreover, we are planning to continuously monitor this project for the next 5 or 10 years to determine any changes in the results upon the improvement of the area’s vitality from increased business and cultural activities and higher diversity, especially to rule out the impact of the pandemic.

Reviewer 3 Report

Sure, here is the grammar and clarity checked version of the text:

The study provides valuable insights into the impact of waterfront renewal on space vitality. However, there are some suggestions for improvement in methodology, as follows:

  • The authors could have provided more detail about the methods used to collect and process the data.
  • The authors could have conducted a more in-depth analysis of the data to identify the specific spatial elements and space vitality indicators that are most affected by waterfront renewal projects.
  • The authors could have compared the results of this study to the results of other studies on waterfront renewal.

Author Response

Reviewer #3 comment 1:

Sure, here is the grammar and clarity checked version of the text:

The study provides valuable insights into the impact of waterfront renewal on space vitality. However, there are some suggestions for improvement in methodology, as follows:

The authors could have provided more detail about the methods used to collect and process the data.

Response:

Thanks a lot for pointing out this. We apologize that the text in section 3.3 Research data collection and treatment is not that clear. It could be found in two sections, one in 3.3.1, line 196-209, already modified as follows, and another in the last line from Table 2, line 187, explaining how to define data information, shown below as well:

(3.3.1 Data collection)…For activity data, we collected two kinds of data respectively in October 2018 and October 2020, which are cellular data from a selected communication operator and POIs on maps. We used monthly spatio-temporal cellular data, statistically overlayed from the instantaneous data from each cell tower, which effectively helps avoid some daily accidental errors. It is worth noting that between 2018 and 2020, during the waterfront regeneration, 24 new cell towers had been installed. After manually deleting these towers, 443 effective cell phone signal towers in the influenced research area remained, each with a precision range of 200 to 500m. In total, 829,539 and 766,388 pieces of cellular data were collected in 2018 and 2020 respectively, including anonymous user IDs of encrypted mobile phones, timestamps, cell tower geographical coordinates, stay length, and stay type. In collecting POI data, we first applied web service access keys and then obtained POI data from maps in October 2018 and October 2020 using Python. Overall, 10,772 and 10,244 pieces of POI data of the influenced research area were selected in 2018 and 2020, respectively (Figure 3).

(Table 2)…Notes: Regarding the population of users: In principle, if users stay at the same location for more than 30 minutes, it is identified as a “stay” and added to the stay length of the user of the cellular tower. According to the monthly stay length, the “stay” type is categorized as “dwelling,” “working,” and “visiting.” If a user stays at a place from 9:00 pm to 8:00 am the next day (9:00 am to 5:00 pm), the “stay” is marked as “dwelling” (“working”). Others are marked as “visiting.” Notes: Regarding the population of users: In principle, if users stay at the same location for more than 30 minutes, it is identified as a “stay” and added to the stay length of the user of the cellular tower. According to the monthly stay length, the “stay” type is categorized as “dwelling,” “working,” and “visiting.” If a user stays at a place from 9:00 pm to 8:00 am the next day (9:00 am to 5:00 pm), the “stay” is marked as “dwelling” (“working”). Others are marked as “visiting.”

Moreover, the text from 3.3.2. Activity data processing: Influenced research area projection method with distance attenuation coefficient, and 3.3.3. Temporal comparison analysis and relationship model, line 211-254, has explained the process of data that is to convert the activity data collected in the influenced research area into each key research unit. In this part, we creatively introduced this distance attenuation coefficient from the gravity model, so that the impact represented by the activity data can decrease exponentially with distance in the influenced research area.

Reviewer #3 comment 2:

The authors could have conducted a more in-depth analysis of the data to identify the specific spatial elements and space vitality indicators that are most affected by waterfront renewal projects.

Response:

Many thanks for pointing out this. Firstly, from the perspective of developing the spatial element evaluation framework, details in section 3.2.1 and table 1, there are two typical factor layers especially in waterfront renewal various from other public spaces, “shoreline breadth” to present “scale”, “proximity to water” to present part of “design”. Then, as a result, especially the factor “proximity to water” is proved to make some difference to the space vitality with more details analysis in sections 4.2 and 5.1. We have revised some text more clearly, line 334-346 as follows:

In general, based on these different space vitality changes, the improvement of spatial elements during renewal construction showed some common features of the trend of the influences, in which performance was broadly “Function” > “Design” > “Accessibility” > “Scale” (Figure 8). In the “Function” dimension, the “commercial and leisure facility density” and “cultural and educational facility density” had the most significant positive influence on space vitality. Social vitality was subject to the relatively significant positive influence of some factors from the “Design” dimension, such as “path density” including walkways, runways, and bicycle ways, and “proximity to water” representing the field of view and closeness to the waterbody, which also a typical and special factor for waterfront various from other public spaces. In terms of “Accessibility”, though the “bus stop coverage” within the pedestrian buffer zone has a relatively positive influence, spatial elements that had weak or even suppressive influences were “path entrance and exit density” and “subway station coverage.”

Reviewer #3 comment 3:

The authors could have compared the results of this study to the results of other studies on waterfront renewal.

Response:

Thanks a lot for your great suggestion. We apologize that the narration in section 5.2 Findings on the elements’ influence compared with prior research is not clear enough. It is a discussion of the four dimensions of special elements in waterfront renewal about the similar and different findings with other prior studies. For example, we have discussed different calculations for “scale” in other public space research and for “proximity to water” in other waterfront research may have different influences on space vitality. We attached the modified text, line 349-382 as follows:

1) The “Function” dimension elements’ improvement has the most significant influence on space vitality in the Maozhou River waterfront. Many waterfront areas are narrow, elongated, and monotonous, and lack the necessary public services and activity spaces. For example, the Dasha River waterfront in the center of the Nanshan district of Shenzhen has been overwhelmed with driveways and has become so narrow that it can only be used as a passageway, and is not a place where people can stay, which is not the same situation discussed with other case studies(Fan et al., 2021). In the Maozhou River waterfront, large venue nodes were built every 500–800m and smaller ones every 200–300m, including public toilets, coffee shops, kid entertainment spaces, sports and leisure facilities, exhibition halls, and science education facilities. We find that these functional places serve as “catalysts” that stimulate a surge in staying behavior and activities, which is similar to findings in other research (Li et al., 2022; Liu et al., 2021).

2) The “Design” dimension elements, such as path density and shoreline proximity to water, have positive influences, while the “Scale” dimension is less effective or even suppresses vitality. If “proximity to water” is calculated based on the field of view and accessibility, its influence is significant, as same as the analysis in the prior research(Liu et al., 2021), but if based on shoreline distance (“shoreline breadth” here), the relationship is weak, which is the same result but in a quite different dimension compared with the prior research (Shi et al., 2021). Shoreline breadth represents the size of the linear public space on the waterfront. If the space is too large (small), it can reduce accessibility and proximity to water (be difficult to realize design and diversity). Shoreline breadth is often an important indicator in urban design or even in urban regulation and control and is worthy of further discussion, as it plays a determining role in the design and positioning of waterfront spaces.

3) The “Accessibility” dimension elements have slightly different effects. Bus stop and parking facility coverage have significant impacts while subway station coverage seems useless, thus contradicting the findings of current public space (streets and waterfront) research in downtown areas from other studies(Tang et al., 2020; Yue et al., 2017). The Maozhou River is in a Shenzhen suburb, where subway station density is quite low compared with other areas in the city. Almost all subway stations in the research area are unsuitable for walking at 500m away from public spaces, and thus fail to increase the vitality of the central research area. Our findings offer a new influence mechanism of waterfront renewal in the suburbs, especially when the focus of existing space renewal gradually moves from the downtown to the suburbs and new towns.